# Unveiling the Synergy of Coupled Gold Nanoparticles and J-Aggregates in Plexcitonic Systems for Enhanced Photochemical Applications

**DOI:** 10.3390/nano14010035

**Published:** 2023-12-22

**Authors:** Alba Jumbo-Nogales, Anish Rao, Adam Olejniczak, Marek Grzelczak, Yury Rakovich

**Affiliations:** 1Centro de Física de Materiales (MPC, CSIC-UPV/EHU), 20018 San Sebastián, Spain; ajumbo001@ikasle.ehu.eus (A.J.-N.); anish.rao@ehu.eus (A.R.); adam.olejniczak@ehu.eus (A.O.); marek.g@csic.es (M.G.); 2Donostia International Physics Center (DIPC), 20018 San Sebastián, Spain; 3Polymers and Materials, Physics, Chemistry and Technology, Chemistry Faculty, University of the Basque Country (UPV/EHU), 20018 San Sebastián, Spain; 4Ikerbasque Basque Foundation for Science, 48013 Bilbao, Spain

**Keywords:** plexcitonic system, metal nanostructures, plasmonic, exciton, J-aggregates, photocatalysis, photodegradation, photostability

## Abstract

Plexcitonic systems based on metal nanostructures and molecular J-aggregates offer an excellent opportunity to explore the intriguing interplay between plasmonic excitations and excitons, offering unique insights into light–matter interactions at the nanoscale. Their potential applications in photocatalysis have prompted a growing interest in both their synthesis and the analysis of their properties. However, in order to construct a high-performing system, it is essential to ensure chemical and spectral compatibility between both components. We present the results of a study into a hybrid system, achieved through the coupling of gold nanobipyramids with organic molecules, and demonstrate the strengthened photochemical properties of such a system in comparison with purely J-aggregates. Our analysis includes the absorbance and photoluminescence characterization of the system, revealing the remarkable plexcitonic interaction and pronounced coupling effect. The absorbance spectroscopy of the hybrid systems enabled the investigation of the coupling strength (g). Additionally, the photoluminescence response of the J-aggregates and coupled systems reveals the impact of the coupling regime. Utilizing fluorescence lifetime imaging microscopy, we established how the photoluminescence lifetime components of the J-aggregates are affected within the plexcitonic system. Finally, to assess the photodegradation of J-aggregates and plexcitonic systems, we conducted a comparative analysis. Our findings reveal that plasmon-enhanced interactions lead to improved photostability in hybrid systems.

## 1. Introduction

In the rapidly advancing field of nanoscience and nanotechnology, the exploration of novel hybrid systems has garnered significant attention, paving the way for innovative applications in photonics, sensing, and energy harvesting [1,2]. Plexcitonic systems are based on the coupling of plasmons and excitons. This interdisciplinary convergence of plasmonics and excitonics holds immense promise for manipulating and harnessing light–matter interactions at the nanoscale [3,4]. To build these systems plasmonic nanoparticles (NPs) can be used to couple to excitons originated from either inorganic [5,6,7,8,9,10] or organic materials [1,11,12,13,14,15]. In the context of organic materials, a plexcitonic system can be established using dyes [16,17] or dye aggregates. These aggregates can be categorized into J-aggregates and H-aggregates, each presenting distinct properties compared to the non-aggregated dyes. These properties arise from the interaction of molecular transition dipole moments that interact upon the formation of molecular chains [18]. Here, J-aggregates are characterized by a sharp and red-shifted absorption and emission response [19], while H-aggregates exhibit a broad and blue-shifted absorption and emission response [20] in comparison to the monomer. Among these, J-aggregates are frequently employed in constructing plexcitonic systems [1,11,12,13,14,15] due to their sharp absorption and emission features as well as small or negligible difference between the absorption and emission maxima (Stokes shift). These distinct features emerge from the exciton coherence length that takes place in the J-aggregates [19,21].

Among the previously described systems, plexcitonic systems which integrate plasmonic excitations in metallic nanoparticles with excitonic states in molecular aggregates stand out as a compelling avenue of research [22,23]. Gold nanoparticles, renowned for their exceptional optical properties related to their localized surface plasmon resonance (LSPR) [24,25], seamlessly combined with J-aggregates, molecular assemblies with distinctive excitonic characteristics [26,27], form the crux of this study. Understanding and manipulating the intricate dynamics within plexcitonic systems not only unravel fundamental insights into the nature of light–matter interactions but also pave the way for the development of advanced nanophotonic devices and efficient energy conversion platforms. In this article, we delve into the contextual significance of studying plexcitonic systems, elucidating their underlying principles, and highlighting their potential impact on emerging technologies.

This paper provides a comprehensive examination of hybrid fluorescent gold bipyrami-dal-like hybrid nanostructures. The chemical synthesis of these gold bipyramids, coupled with fluorescent J-aggregates utilizing a cost-effective and affordable non-cyanine dye, is detailed along with an exploration of their photophysical properties. Another important aspect is the use of electrostatic interactions to reliably form plexcitonic systems. For this, interaction with negatively charged molecules was required and it was crucial to develop a surface functionalization strategy to make BPs positively charged by a ligand exchange process. We not only demonstrate a new plexcitonic system comprising gold bipyramids (AuBPs) and S2275 dye J-aggregates but also show the impact of this coupling in enhancing the photostability of the J-aggregates and expanding the absorption spectral range of the plexcitonic system when compared to bare J-aggregates. The advantages of our system arise not only from the unique characteristics of the components but also from their coupling. For instance, the absorption and emission response of J-aggregates in the NIR region renders them suitable for biological applications. Furthermore, Au BPs are anisotropic sharp gold nanostructures that display intriguing characteristics, which are attributed to the potent electric field generated at their edges and tips. They exhibit narrow and intense LSPR response and have only recently gained attention for building plexcitonic systems. To the best of our knowledge, there exist a limited number of studies involving AuBPs/J-aggregates of cyanine dyes [28,29]. However, it is worth noting that despite the common use of TDBC in strong coupling experiments [30], its stability in J-aggregate morphology is rather compromised, as is the case with most cyanine dyes. Most notably, we demonstrate the fabrication of the plexcitonic system in a solution state, enabling the simultaneous production of multiple homogeneously coupled AuBP/J-aggregates nanosystems.

## 2. Materials and Methods

### 2.1. Materials

The chosen dye for this work was 5-Chloro-2-[3-[5-chloro-3-(4-sulfobutyl)-3H-benzothi-azol-2-ylidene]-propenyl]-3-(4-sulfobutyl)-benzothiazol-3-ium hydroxide, inner salt, triethylammonium salt (S2275 dye) and was purchased from FEW Chemicals (Wolfen, Sachsen-Anhalt, Germany). To form the J-aggregates, we used sodium chloride (NaCl) from Sigma Aldrich (St. Louis, MO, USA). For the particle stabilization, we used cetyltrimethylammonium bromide (CTAB), (11-Mercaptoundecyl)trimethyl-ammonium bromide (TMA), and poly(diallyldimethylammonium chloride) (PDDA) that were purchased from Sigma Aldrich and 11-Mercaptoundecane-1-sulfonic acid sodium salt (MUS) that was purchased from Prochimia (Gdynia, Poland). Borosilicate cover glass (thickness 0.13–0.16 mm) was used to prepare solid-state samples and was purchased from VWR International (Radnor, PA, USA).

### 2.2. Formation of J-Aggregates

The S2275 dye presents its monomeric form in a methanol solution with an absorbance maximum of 566 nm (see Appendix A). However, to observe the formation of J-aggregates, a NaCl aqueous solution is required. The formation of J-aggregates results in the appearance of a sharp absorbance J-band at 650 nm (see Appendix A). The intensity of the J-band increases with increasing concentrations of NaCl.

### 2.3. Synthesis of AuBPs

Gold nanobipyramids were synthesized by growing gold pentatwinned seeds [31] and stored in CTAB 15 mM solution.

### 2.4. Ligand Exchange Au-TMA

First, the AuBPs were functionalized using TMA to obtain positively charged particles. To begin with, AuBPs-CTAB were centrifuged (10 mL, [Au] = 0.5 mM, 1 mM CTAB) and redispersed in Milli-Q water (8 mL). To this solution, TMA (2 mg/mL, 2mL) was added dropwise under magnetic stirring. After incubation for 1 h, the functionalized particles were centrifuged and redispersed in water. This purification step was repeated two times.

### 2.5. Ligand Exchange Au-MUS

The influence of electrostatic interactions on the formation of the plexcitonic system was tested functionalizing the AuBPs with a negatively charged molecule. Here, MUS and a functionalization strategy similar to what was used for TMA was employed.

### 2.6. Hybrid Systems

The synthesis of gold nanoparticles–J-aggregates hybrid systems was carried out in solution. The hybrids were prepared by adding dye (1 mM, 70 μL) to AuBPs-TMA ([Au^0^] = 0.5 mM, 2 mL). Subsequently, we added NaCl (4 M, 25 μL) under magnetic stirring to initiate the formation of J-aggregates. Within 10 min, we observed the spectroscopic signatures indicating the formation of a plexcitonic system, i.e., the formation of a doublet structure near the exciton and plasmon resonance position. A similar protocol was followed to study the formation of a plexcitonic system with AuBPs-TMA and AuBPs-MUS.

### 2.7. Substrate Preparation

Also, a J-aggregate layer was deposited on a borosilicate cover glass for performing photophysical measurements. The cover glass was cleaned using acetone and ethanol and subsequently functionalized with a positively charged polyelectrolyte, i.e., PDDA. To do this, the cleaned glass was covered with a thin layer of PDDA solution (1 mg/mL) and left undisturbed for 1 h. The cover glass was then washed with water to remove the excess of PDDA. Finally, 100 μL drop of J-aggregates solution was placed over the functionalized glass surface for 1 day. Subsequently, the excess solution was removed with water, and the cover glass was dried under nitrogen flow. The same procedure was followed to deposit the hybrid systems on the glass substrate.

### 2.8. Characterization

The J-aggregates, bare AuBPs, and hybrid solutions were characterized using a Cary 3500 spectrophotometer to obtain their absorbance (UV-Vis-NIR) spectra. The emission spectra were measured through Cary Eclipse Fluorescence Spectrophotometer (Agilent Technologies, Santa Clara, CA, USA). For the photoluminescence (PL) decays and Fluorescence Lifetime Imaging Microscopy (FLIM) analysis, we used a time-resolved confocal fluorescence microscope system MicroTime 200 and a laser excitation of 485 nm. The PL response of monomer and J-aggregates was separated using two filters: one centered at 550 nm to obtain just monomeric response and the second at 650 nm for J-aggregates. The PL lifetime values were estimated by fitting the photoluminescence decay curves with SymPhoTime 64 software. It must be pointed out that all these measurements were performed under ambient conditions.

Also, in the case of J-aggregates and hybrid layers, the absorbance spectra allowed us to monitor the molecule’s J-aggregation and their response to the NaCl concentration in the original solution. The UV-Vis-NIR response of glass substrates containing J-aggregates and plexcitonic systems was measured using a Varian Cary 50 spectrophotometer. Optical images of liquid J-aggregates and hybrid AuBPs-J-aggregates samples were obtained using an optical microscope integrated with a confocal Raman microscopy setup (Alpha300, WITec, Ulm, Germany) from a droplet of a liquid sample deposited on a microscope glass slide. Objectives of 20× (0.4 NA) and 100× (0.8 NA) were used for imaging J-aggregates and hybrids, respectively.

For light irradiation experiments, we employed a G2V pico solar simulator, tuning the light wavelength to the range of 650 to 800 nm with 90.4 mW power.

## 3. Results and Discussion

### 3.1. J-Aggregates

The overall objective of the present work is to study the formation of plexcitonic systems and demonstrate the effect of plexcitonic coupling on the photophysical and photochemical properties of S2275 dye. It is well established that an increase in ionic strength, achieved through the addition of NaCl, induces the self-assembly of S2275 dye molecules into J-aggregates [32,33]. Figure 1a (red spectrum) gives the UV-Vis-NIR response of S2275 dye showing an intense band in the 400–500 nm region. As the concentration of NaCl in the solution increases, a distinct band with an absorbance maximum at 650 nm emerges, signifying the J-aggregate response. Figure 1a also shows the simultaneous reduction in the UV-Vis-NIR response from dye monomers (indicated by a pink arrow) and growing response from J-aggregates (denoted by a blue arrow) with the increasing ionic strength of the solution. The response of J-aggregates (absorption at 650 nm) demonstrates a notable increase as the NaCl concentration rises (Figure 1b). This study allows us to find the minimum concentration of NaCl (∼97.4 mM) that is required to form J-aggregates of S2275 dye (10 μM). For more details, see Appendix A.

### 3.2. Plexcitonic Systems

After demonstrating the formation of J-aggregates, we moved on to the second component of our plexcitonic system, i.e., plasmonic nanoparticles. In the present study, we have chosen AuBPs because they show large optical cross-sections resulting in large local electric field enhancements. Also, they present narrower absorbance line widths, better shape and size uniformity, and higher refractive index sensitivity [34] than other gold nanoparticle morphologies.

The second component of our plexcitonic system is gold bipyramids. Since the present work aims to establish plexcitonic systems using S2275 J-aggregates, it is crucial to select AuBPs with matching spectral properties. With this in mind, we used established literature protocols to synthesize AuBPs with LSPR maxima at 680 nm (see Figure 1c), closely matching the UV-Vis-NIR maxima of J-aggregates. Figure 1d shows the TEM images of the prepared AuBPs having an average length of 62 ± 3 nm and width of 34 ± 2 nm. Next, the zeta (ζ) potential of CTAB-coated AuBPs was measured to be +52.3 ± 20 mV, indicating a high positive surface charge. Since J-aggregates formed using S2275 dyes demonstrate a negative surface charge [35,36] (discussed vide infra), we hypothesized exploiting electrostatic interactions to govern the formation of the plexcitonic system. With this in mind, we added 35 μM of dye to [Au^0^] = 0.5 mM AuBPs in CTAB solution in the presence of 50 mM NaCl. Appendix A shows the additive spectra of J-aggregates, along with the UV-Vis-NIR response of AuBPs, indicating the absence of plexcitonic coupling. The lack of spectroscopic signatures corresponding to the plexcitonic system in Appendix A indicates the inability of AuBPs-CTAB and J-aggregates to form hybrids. Furthermore, both the J-aggregates and AuBPs could be separated under centrifugation, confirming the absence of any plexcitonic system. This absence of interaction is possibly due to the presence of free CTAB micelles in the solution. To circumvent this challenge, a ligand exchange reaction was employed to replace the native CTAB on the AuBPs surface with positively charged TMA. Here, TMA was chosen due to its strong affinity toward Au through strong Au-S bonds [37] and its ability to impart permanent positive charge to AuBPs through quaternary ammonium headgroups. Figure 1c demonstrates the UV-Vis-NIR spectra of AuBPs before and after the ligand exchange process. The slight blue shift of the LSPR peak after TMA functionalization [38] is due to the change in the refractive index of the medium from 1.46 (in CTAB) to 1.33 (in water) [39]. Crucially, ζ-potential measurements confirm the positive charge on AuBPs following TMA functionalization (ζ-potential: +47.9 ± 19 mV). Functionalizing AuBPs with TMA circumvents the difficulties due to the presence of free micelles in the solution. Consequently, 35 μM of S2275 was introduced to [Au^0^] = 0.5 mM AuBP-TMA in the presence of 50 mM NaCl, resulting in the formation of a plexcitonic system. Figure 1e shows the optical and TEM images of the hybrid system at three different length scales: a macroscale view of the solution containing the plexcitonic system, an optical microscope image (microscale) of AuBPs comprising the plexcitonic system, and a TEM image (nanoscale) showcasing the particles embedded in a net of aggregated dye. Finally, Figure 1f shows the spectroscopic signatures of the plexcitonic system in the UV-Vis-NIR spectrum, i.e., the appearance of two new peaks (at 650 nm, shown in purple). The origin of this doublet is due to the formation of new light–matter hybrid states [1]. Additionally, Figure 1f shows the UV-Vis-NIR spectrum of plasmonic (shown in pink) and excitonic (shown in blue) components comprising the plexcitonic system.

Having successfully demonstrated the formation of a plexcitonic system, we aimed to gain quantitative insights into the underlying mechanism guiding its formation. We hypothesize that electrostatic attractions govern the creation of plexcitonic systems. To test this hypothesis more thoroughly, we prepared AuBPs and exchanged the native CTAB with a negatively charged ligand (MUS). Subsequently, with AuBPs-MUS ready, we introduced 70 μL of S2275 dye into a solution comprising [Au^0^] = 0.5 mM of AuBPs-MUS and 50 mM NaCl. Over the course of 2 h, we monitored the UV-Vis-NIR spectrum of the solution. Notably, no spectroscopic signatures corresponding to the formation of a plexcitonic system were observed even after 2 h. The variations in the UV-Vis-NIR spectrum of the system (shown in Figure 2a) are due to the screening of electrostatic repulsions and ultimately, the aggregation of AuBP-MUS at such high salt concentrations [40]. Contrastingly, upon repeating the experiment using AuBPs-TMA (Figure 2b), we observed a red shift after the addition of dye and the almost immediate appearance of a doublet structure in the UV-Vis-NIR spectrum within 3 min. This dip in the UV-Vis-NIR spectrum, signaling the formation of a plexcitonic system, continued to intensify over the course of 2 h. It is important to note that we observed distinctly different kinetics for the formation of J-aggregates in the presence and absence of AuBPs-TMA (Figure 2c), where J-aggregates formed within 2 min in the absence of AuBPs-TMA. Furthermore, the conditions for inducing J-aggregate formation in the presence and absence of AuBPs-TMA are distinctly different, too. Most notably, the concentration of NaCl to induce the formation of J-aggregates in the absence of AuBPs-TMA is 100 mM (Figure 2d), i.e., twice of what is used during the formation of the plexcitonic system. This reduced need for NaCl indicates the active participation of AuBPs-TMA in the plexcitonic system formation.

Following the successful demonstration of a plexcitonic system formation, our focus shifted to studying the nature and strength of interactions within these hybrid structures. With this in mind, we tuned the LSPR in the plasmonic systems comprising the plexcitonic hybrids. Specifically, we prepared six AuBPs samples with LSPR maxima ranging from 612 to 700 nm and subsequently functionalized them with TMA. This variation in the position of plasmonic LSPR peaks allowed us to easily adjust the tuning in our plexcitonic system. This plasmonic range variation ensures thorough testing of the coupling strength in our plexcitonic system. Figure 3a demonstrates the successful formation of plexcitonic systems with these six AuBPs having differing spectral mismatches.

The analysis of the coupling strength in the plexcitonic systems requires the estimation of the Rabi-splitting energy ($\Omega_{R}$). The Rabi-splitting energy was calculated by fitting the dispersion curves (shown in Figure 3b) using Equation (Equation 1) [41]. The value obtained for $\Omega_{R}$ was 130 meV.
(1)$$E_{\pm }=\frac{1}{2}\left(E_{ex}+E_{SP}\right)\pm \sqrt{\Omega_{R}^{2}+\frac{1}{4}{\left(E_{SP}-E_{ex}\right)}^{2}}$$
In this expression, $E_{\pm }$ refers to the upper and lower plexcitonic resonances, $E_{ex}$ is the exciton energy given by the J-band spectral position, and $E_{SP}$ is the plasmonic response. Following the estimation of $\Omega_{R}$, the coupling strength (*g*) of the plexcitonic system was estimated using the following relation [15,41]: (2)$$\Omega_{R}=\sqrt{4g^{2}-\frac{1}{4}{\left(\Gamma_{SP}-\Gamma_{exc}\right)}^{2}}$$
where $\Gamma_{SP}$ (0.179 eV) and $\Gamma_{ex}$ (0.0555 eV) represent the line widths of plasmon and exciton absorbance responses, using established literature protocols [42]. The obtained coupling strength for this system is 72 meV, and from this estimated value, it is possible to evaluate the coupling regime of the plexcitonic system. The first condition to be satisfied involves the comparison of the coupling strength and the exciton and plasmon decay rates: $g>1/2(\Gamma_{SP}-\Gamma_{ex})$ [15]. Also, since it is considered that at least one complete Rabi oscillation takes place for the strongly coupled system, the coupling strength should accomplish: $g>1/4(\Gamma_{SP}+\Gamma_{ex})$ [15]. In our case, both criteria are fulfilled as well as the relation: $4g^{2}\gg \Gamma_{SP}\Gamma_{ex}$ [12,13,43,44]. To conclude this part of the analysis, we present the Hopfield coefficients [45] estimation for the established plexcitonic system: for the Upper Resonance Branch (URB) in Figure 3c and the Lower Resonance Branch (LRB) in Figure 3d. Through these coefficients, it is possible to quantify the exciton and plasmon fractions present in the plexcitonic system as a function of varying the LSPR energies. According to this analysis, the excitonic fraction decreases with transition energy from ≈83% to 14% in the Upper Resonance Branch, and the plasmonic fraction evolves from 16% to 86%.

One of the primary objectives of this study was to investigate the impact of plexcitonic coupling on the photophysical and photochemical characteristics of J-aggregates. Some of these molecular chains are characterized for presenting two lifetimes components: a fast component interpreted as the result of the excitation quenching by the exciton–exciton annihilation of the excited J-aggregate and a slow component due to the excited aggregate molecule [46,47]. This multiexponential PL decay in J-aggregates has its origin in the exciton coherence length in the molecular chain [21,48]. Specifically, we employed both PL imaging and FLIM techniques for the complete photophysical characterization of our system. These techniques allowed us to thoroughly assess the impact of plexcitonic coupling on the PL intensity, the lifetime components, and the spatial distribution of PL lifetimes of J-aggregates. Firstly, the PL response from J-aggregates was analyzed using a dried sample on a glass substrate. Figure 4a shows the PL intensity map of the glass substrate coated with J-aggregates. In the colormap, the PL response from J-aggregates is shown in more intense red colors. Next, we extracted the average PL lifetime decay of J-aggregates in the mapped region. Figure 4c shows the biexponential PL decay belonging to J-aggregates with lifetime components $\tau_{1}$ = 0.3 ns, and $\tau_{2}$ = 2.3 ns, as it commonly observed [14,42,48]. Employing the FLIM analysis, we visualized the distribution of these lifetimes in the mapped area. Figure 4e shows the distribution of both lifetimes, demonstrating the prevalence of the shorter lifetime component. In contrast, we observe a weaker PL response from the plexcitonic system (see Figure 4b, which was previously observed in gold nanoparticles’ interaction with emitters [49,50]). More specifically, we observe significantly lower photoluminescence intensity in the plexcitonic system (reduced by a factor of 100) compared to J-aggregates, where the PL intensity is around 1000. Finally, we used the PL decay belonging to the plexcitonic systems (shown in Figure 4d) to map the distribution of lifetimes in the imaged area. Here too, we obtained a biexponential PL decay with lifetime components $\tau_{1}$ = 0.3 ns and $\tau_{2}$ = 1.7 ns. Interestingly, we observe a reduction in the longer lifetime component and an increment in the shorter one. An important point to note here is that the FLIM analysis of the plexcitonic system, depicted in Figure 4f, reveals comparable amplitudes for both lifetime components.

The change in lifetime components can be attributed to the J-aggregates’ chain size variation, since a decreased lifetime was reported for an increased J-aggregates length [51]. With this, our hypothesis lies on the J-aggregates’ size increase when they interact with particles to explain the measured shorter lifetime. Also, we can point out that J-aggregates formation requires less salt concentration when using gold particle solution, so AuBPs could act as aggregation centers for the molecules due to their charge interaction.

Finally, we investigate the impact of plexcitonic coupling on the photochemical characteristics of J-aggregates. More specifically, we examine the photodegradation of dye J-aggregates both in the presence and absence of AuBPs under illumination with a solar simulator (wavelength range: 650–800 nm). For more details, see Appendix A. This examination enables us to secure the stability and conservation of electronic states and optical transitions within hybrid nanostructures against the influence of the surrounding environment [11]. The variations in the UV-Vis-NIR spectrum of plexcitonic systems under illumination are shown in Figure 5a. Initially, we observe negligible variations in the UV-Vis-NIR response for ∼2 h, after which noticeable red shifts become apparent in the presented spectrum. This red shift is indicative of the aggregation of AuBPs in the solution, which is confirmed by the absence of spectroscopic signals from AuBPs within 4 h (yellow spectrum in Figure 5a). The J-aggregates’ spectra evolution can be seen in Figure 5b, where the J-band decrease is noticeable from the first spectra taken after illumination. This fast intensity reduction in the J-aggregates response with light exposure leads us to conclude that molecules are photodegraded. This drastic difference in the stability of J-aggregates in the presence and absence of AuBPs is clearly demonstrated in Figure 5c. Here, we track the photostability of the plexcitonic system by monitoring the dip at 650 nm and the photostability of J-aggregates by monitoring the peak at 650 nm. In this context, J-aggregates demonstrated photostability in the presence of plexcitonic coupling with AuBPs for ∼2 h, while 67% of J-aggregates underwent photobleaching in the same time frame. This exceptional resistance to photodegradation of our plexcitonic system (for ∼2 h) is directly attributed to the colloidal stability given by the AuBPs interaction with J-aggregates. Therefore, we provide compelling evidence that the plexcitonic system exhibits superior photostability when compared to bare J-aggregates. At the same time, the presented system is limited by the particles aggregation due to electrostatic interactions between negatively charged dye and positively charged particles. Furthermore, photothermal effects can also come into play under prolonged exposure to light and can contribute to the instability of the system, as seen in Figure 5a spectra after two hours of light exposure. This issue can be resolved by appropriately varying the surface chemistry of the AuBPs, where the interaction between the NPs and the dye can be established along with the enhanced colloidal stability of the plexcitonic system.

## 4. Conclusions

A new plexcitonic system was established using plasmonic nanostructures and J-aggregates. A competitive optical characterization by the system absorbance and PL response in the NIR spectral region was performed. Using these data, we were able to determine the Rabi-splitting energy and investigate the coupling regime through the estimated coupling strength parameter. There exist several parameters to distinguish the coupling regimes, and according to most of them, the system interacts in the strong coupling regime. Finally, we investigated the changes in the photophysical and photochemical properties through its photoluminescence response and photodegradation. The presented hybrid systems show improved properties in comparison to bare J-aggregates like enhanced photostability and widened absorption spectral range. The plexcitonic system’s emergent properties are due to the two hybrid states’ formation; they make these hybrid systems suitable for different applications. In photocatalysis, there are already some coupled systems that have been applied, like Au NPs coupled to a Fabry–Pérot nanocavity that resulted in enhanced water splitting [5,52] and water oxidation [6].

The limitations of these plexcitonic systems are drawn by the nature of their components. These emerge from the aggregation of the Au BPs in the presence of oppositely charged dye monomer as well as the sensitivity of the plexcitonic system to photothermal effects. However, we presume that it is possible to overcome these limitations by appropriately varying the surface chemistry of the AuBPs, where the interaction between the NPs and the dye can be established along with the enhanced colloidal stability of plexcitonic system.

## Figures and Tables

**Figure 1 nanomaterials-14-00035-f001:**
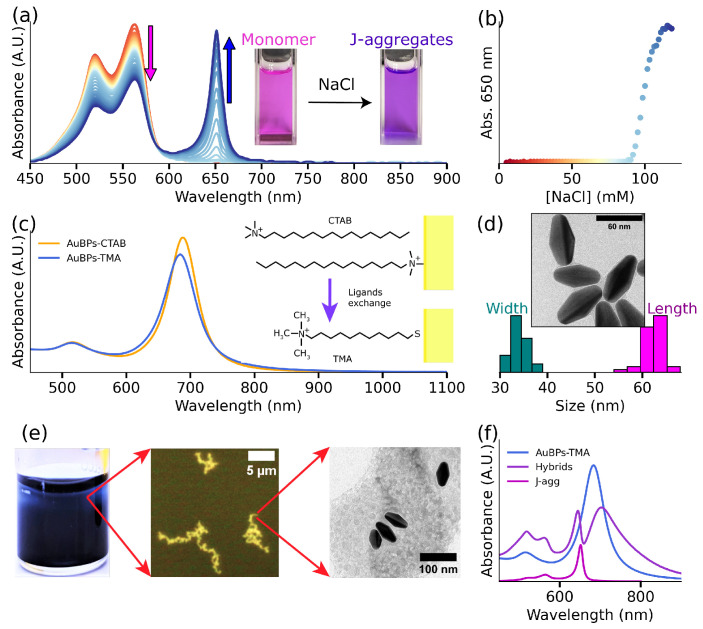
J-aggregates and gold nanoparticles characterization. (**a**) J-aggregates formation under NaCl effect, the absorbance spectra show the influence of NaCl molar concentration in the solution in the J-aggregates formation: an increasing J-band response and decreasing monomeric one. (**b**) J-band (650 nm) evolution as a function of NaCl concentration in the solution. (**c**) Ligands exchange: absorbance response of AuBPs in CTAB solution, after synthesis, (orange line) and after TMA functionalization (blue line). The right inset shows the molecules’ structure on the gold surface. (**d**) TEM image of synthesized gold nanoparticles and the obtained TEM size statistics. (**e**) Hybrid systems at different scales: solution (macroscopic), optical image (micrometric), and TEM image (nanometric). (**f**) Hybrids formation: absorbance spectra of AuBPs (blue), J-aggregates (magenta), and hybrid systems (purple).

**Figure 2 nanomaterials-14-00035-f002:**
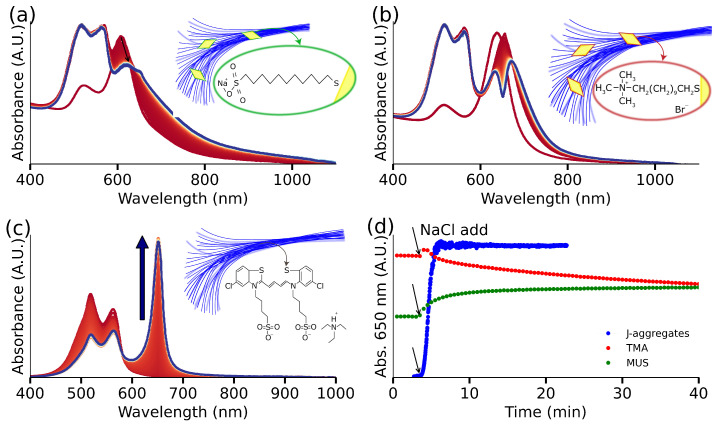
J-aggregates formation, their interaction with functionalized gold nanoparticles: with TMA (positive charge) and MUS (negative charge), and J-band evolution for hybrids’ solution and J-aggregates solution. (**a**) AuBPs functionalized with MUS evolution after J-aggregates formation; the inset image shows a hypothetical structure of the system: particles over a net from the aggregated dye. (**b**) AuBPs functionalized with TMA evolution after J-aggregates formation, the inset exhibits a hypothetical structure of the obtained plexcitonic system: the aggregated dye forms a net, acting as a particle trap. (**c**) NaCl-triggered J-aggregates form in solution after the addition of NaCl: the J-band grows with time, and monomer response decreases; the inset shows a hypothetical structure of aggregated dye elongated morphology. (**d**) The J-band interaction was monitored in three cases: for the J-aggregates alone, we obtain the J-band growth, for the plexcitonic system, it shows the dip evolution due to the doublet structure and in the case of MUS functionalized particles, J-band response and particle aggregation is observed.

**Figure 3 nanomaterials-14-00035-f003:**
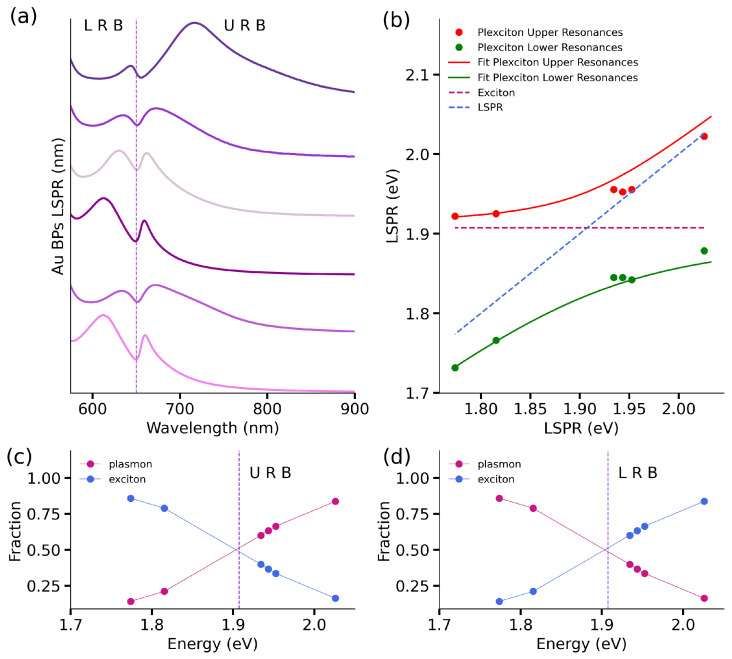
Hybrids of functionalized AuBPs with TMA and J-aggregates absorbance characterization. (**a**) The absorbance spectra of the plexcitonic systems show the two branches originated from the two new hybrid states in the system. Spectra are arranged (from top to bottom) in decreasing order of LSPR wavelengths (612–700 nm). (**b**) The dispersion curve was obtained by plotting the plexcitonic system’s upper (red dots) and lower resonances (green dots) in comparison to the J-aggregates excitonic response (magenta dotted line) and the plasmonic response (blue dotted line). The fitted curves obtained by applying Equation (Equation 1) to the plexcitonic upper and lower resonances are represented by the red and green continuous lines. The estimated Hopfield coefficients show the plasmon (magenta dots) and exciton (blue dots) fractions in the plexcitonic system, according to the estimated Rabi-splitting energy. In (**c**) the fractions corresponding to the Upper Resonances Branch and in (**d**) the Lower Resonances Branch.

**Figure 4 nanomaterials-14-00035-f004:**
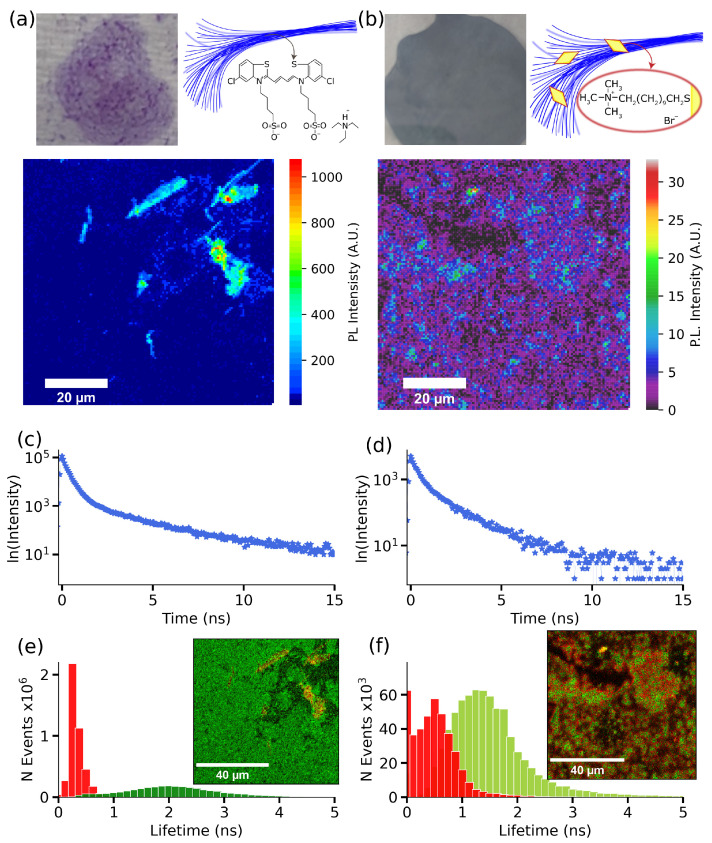
Photoluminescence characterization of deposited S2275 J-aggregates and hybrids systems on substrates. (**a**) On the top left side, we have the image of the deposited J-aggregates over a glass substrate and a hypothetical structure of the aggregates’ elongated morphology on the right side. The lower image corresponds to the PL intensity image obtained after scanning an 80 × 80 μm zone of the J-aggregates deposited layer on the glass substrate. (**b**) On the top left side, we have the plexcitonic system layer deposited on a glass substrate and a hypothetical structure of the AuBPs on the aggregates’ elongated morphology on the right side. The lower image corresponds to the photoluminescence intensity colormap of the sample. (**c**) The average PL decay was obtained from the photoluminescence colormap for J-aggregates. (**d**) The photoluminescence decay corresponds to the colormap PL intensity for the hybrid system. (**e**) The J-aggregates lifetimes histogram obtained from the FLIM analysis applied to the PL colormap; the inset shows the lifetime colormap according to the estimated lifetime components. (**f**) The plexcitonic system lifetimes histogram was obtained from the FLIM analysis applied to the PL colormap, and the inset image shows the lifetime colormap according to the estimated lifetime components.

**Figure 5 nanomaterials-14-00035-f005:**
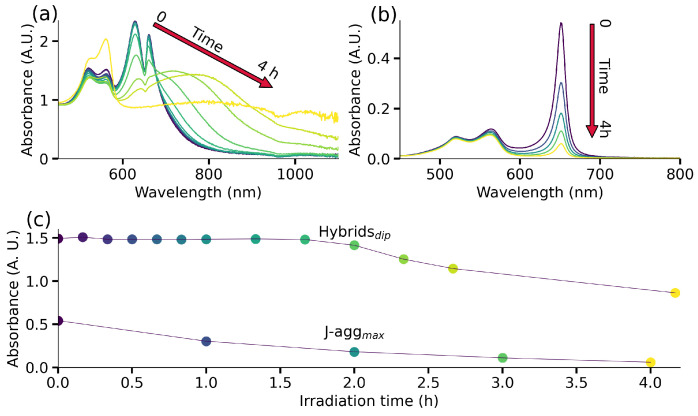
Photostability of plexcitonic system. (**a**) The time-dependent UV-Vis-NIR spectra of plexcitonic system show invariance of the spectral signature, which is followed by a gradual broadening of LSPR due to aggregation. (**b**) The absorbance response of J-aggregates shows degradation under light irradiation. (**c**) Evolution of the dip between the doublet structure of the plexcitonic system to asses the system colloidal stability under light irradiation and the J-band intensity to quantify the J-aggregates’ degradation.

## Data Availability

Data are contained within the article and Appendix A.

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
