# Peer review of "Unveiling the Synergy of Coupled Gold Nanoparticles and J-Aggregates in Plexcitonic Systems for Enhanced Photochemical Applications"

_nanomaterials, 2023, doi:10.3390/nano14010035_

Round 1
Reviewer 1 Report
Comments and Suggestions for Authors
1. The Introduction section does not provide sufficient background information or context to help readers understand the significance of your research. We suggest that you consider adding more detail to help readers better understand the importance of your work.
2. There are several instances where the descriptions were unclear, and some of the tables and figures contained errors (For example, Figures 2 is not available). Please review the manuscript carefully and make the necessary corrections to ensure that your study's presentation is clear and accurate.
3. Many related studies have been reported before, please elaborate on the advantages of the methods or materials in this study compared to previous studies.
4. It remains unclear what the potential value of your study is. We suggest that you articulate this aspect more clearly and persuasively in the manuscript.
5. It would be helpful if you discussed the limitations of your study more explicitly.
6. We suggest that the conclusion section needs to be presented in more detail so that readers can clearly understand the goals and results of the study.
Comments on the Quality of English Language
The English is okay.
Reviewer 2 Report
Comments and Suggestions for Authors
The author successfully established a new plexcitonic system using plasmonic nanostructures and J-aggregates. They provide compelling evidence that the plexcitonic system exhibits superior photostability when compared to bare J-aggregates. Some revisions should be completed before it’s accepted by the journal of Nanomaterials.
1. In the part of “2.2. Formation of J-Aggregates”, please make clear the “Figure ??”. Also, need to check more “Figure ??”.
2. Please give more discussion about J-aggregate and H-aggregate in the Introduction.
3. Please explain the overlapping areas between Absorption and PL spectra in Figure S1 and S2.
4. On page 5 of SI, Figure S7 should be revised as Figure S8.
5. Adding the absorption spectrum of the dye in the solid state and comparing all absorption spectra of dye solid, dye solution with or without J-aggregate, hybrid system.
6. Why peak intensity of the hybrid system in solution show a large difference in Figure S6b (green line) and S8?
7. Why t1 = 0.3 ns, and t2 = 2.3 ns are so different in the transient PL measurement?
8. Please give more outlook about this plexcitonic system for future potential application.
Reviewer 3 Report
Comments and Suggestions for Authors
The evaluated manuscript deals with the description of innovative plexitonic systmes based on aggregated dyes molecules and Gold nanopyramids. The paper is well organized, the language is clear and concise and the figure are of high quality. Results are well supported by experimental data and properly discussed.
Manuscript could be accepted for publication after minor revisions, as listed below.
Minor Point
1) Experimental Section should be presented as a unique paragraph and not as a bullet-point list. Moreover, more details are required in order to allow the reproducibility of the results.
2) Please check the reference to figures. It appears a "??" in the manuscript.
3) In the conclusions, authors should better clarify which are the possible impact of the work, with a specific focus on photocatalysis (mentioned in the abstract).
Round 2
Reviewer 2 Report
Comments and Suggestions for Authors
can be accepted without further revision.